# Gambierol Blocks a K^+^ Current Fraction without Affecting Catecholamine Release in Rat Fetal Adrenomedullary Cultured Chromaffin Cells

**DOI:** 10.3390/toxins14040254

**Published:** 2022-04-02

**Authors:** Evelyne Benoit, Sébastien Schlumberger, Jordi Molgó, Makoto Sasaki, Haruhiko Fuwa, Roland Bournaud

**Affiliations:** 1Service d’Ingénierie Moléculaire pour la Santé (SIMoS), Département Médicaments et Technologies pour la Santé (DMTS), Institut des Sciences du Vivant Frédéric Joliot, Université Paris-Saclay, CEA, INRAE, ERL CNRS 9004, F-91191 Gif-sur-Yvette, France; evelyne.benoit@cea.fr; 2CNRS, Laboratoire de Neurobiologie Cellulaire et Moléculaire-UPR 9040, F-91198 Gif-sur-Yvette, France; sebastienschlumberger@hotmail.com; 3Graduate School of Life Sciences, Tohoku University, Sendai 980-8577, Japan; masasaki@tohoku.ac.jp; 4Department of Applied Chemistry, Faculty of Science and Engineering, Chuo University, Tokyo 112-8551, Japan; hfuwa.50m@g.chuo-u.ac.jp

**Keywords:** fetal adrenomedullary chromaffin cell, gambierol, potassium currents, calcium-activated K^+^ channels, ATP-sensitive K^+^ channels, catecholamine release

## Abstract

Gambierol inhibits voltage-gated K^+^ (K_V_) channels in various excitable and non-excitable cells. The purpose of this work was to study the effects of gambierol on single rat fetal (F19–F20) adrenomedullary cultured chromaffin cells. These excitable cells have different types of K_V_ channels and release catecholamines. Perforated whole-cell voltage-clamp recordings revealed that gambierol (100 nM) blocked only a fraction of the total outward K^+^ current and slowed the kinetics of K^+^ current activation. The use of selective channel blockers disclosed that gambierol did not affect calcium-activated K^+^ (K_Ca_) and ATP-sensitive K^+^ (K_ATP_) channels. The gambierol concentration necessary to inhibit 50% of the K^+^ current-component sensitive to the polyether (IC_50_) was 5.8 nM. Simultaneous whole-cell current-clamp and single-cell amperometry recordings revealed that gambierol did not modify the membrane potential following 11s depolarizing current-steps, in both quiescent and active cells displaying repetitive firing of action potentials, and it did not increase the number of exocytotic catecholamine release events, with respect to controls. The subsequent addition of apamin and iberiotoxin, which selectively block the K_Ca_ channels, both depolarized the membrane and enhanced by 2.7 and 3.5-fold the exocytotic event frequency in quiescent and active cells, respectively. These results highlight the important modulatory role played by K_Ca_ channels in the control of exocytosis from fetal (F19–F20) adrenomedullary chromaffin cells.

## 1. Introduction

Marine dinoflagellates are the source of a well-documented and distinctive group of bioactive polycyclic ether natural products, showing numerous associated ether rings in a ladder shape. These ladder-shaped polyethers are mainly found in marine microorganisms and are considered as secondary dinoflagellate metabolites constituting a rich source of complex compounds (for reviews see [1,2,3]). The demanding steps for their isolation and purification, and the small quantities obtained, have severely limited their structural and bioactivity characterization. Fortunately, some of these complex polyether molecules have been amenable to total organic chemical synthesis (for reviews see [4,5,6]). Among these marine polyethers, gambierol and analogs have been synthesized by means of diverse syntheses strategies [7,8,9,10], allowing detailed studies of their actions in a number of biology models.

Gambierol, distinguished by a transfused octacyclic polyether core (Figure 1), was the first toxin isolated and characterized from cultured *Gambierdiscus toxicus* dinoflagellates gathered from the Rangiroa peninsula in French Polynesia [11,12]. The genus’ *Gambierdiscus* and *Fukuyoa* produce numerous ladder polycyclic ether compounds, including the well-identified ciguatoxins. This family of toxins is responsible for ciguatera poisoning, a seafood-borne disease resulting from the consumption of fish from tropical or temperate waters, or marine invertebrates that have bioaccumulated ciguatoxins (reviewed in [13,14,15,16]). It has been suggested that gambierol participates in ciguatera fish poisoning, but direct proof for this assumption has not yet been given, and to the best of our knowledge, the polyether toxin has not yet been identified in ciguateric fish, maybe because it is present at very low concentrations.

Nanomolar concentrations of gambierol and analogs inhibit voltage-gated K^+^ (K_V_) channels in various excitable cells [17,18,19], including cells expressing K_V_1.1–K_V_1.5 channels [20,21,22] or K_V_3.1 channels [23], and in frog and mouse motor nerve terminals [24,25].

Adrenomedullary chromaffin cells are known to generate action potentials [26], and to display voltage-gated Na^+^ (Na_V_) channels that are sensitive to tetrodotoxin (TTX) [27], and they are also involved in regulating the firing rate of action potentials [28]. The abundant diversity of K_V_ channels in chromaffin cells highlights their fundamental role in the control of the electrical properties of these cells, including the speed of action potential repolarization, the duration of the after-hyperpolarization, the firing rate, and the resting membrane potential (reviewed in [29]). In addition, individual chromaffin cells, depending on the animal species and stage of development, express distinct subtypes of voltage-gated Ca^2+^ (Ca_V_) channels, including low-voltage-activated T-type (Ca_V_3) channels [30], high-voltage-activated channels comprising L-type (Ca_V_1.2 and Ca_V_1.3), P/Q-type (Cav2.1), N-type (Ca_V_2.2) [31] and R-type (Ca_V_2.3) channels (for reviews see [29,32,33]).

The adrenomedullary chromaffin cells secrete catecholamines in response to various stressors, including acute hypoxia [34]. In adult mammals, the catecholamine secretion is triggered by the sympathetic nervous system that supplies, via the splanchnic nerve, the cholinergic innervation to the cells. In the perinatal period, the splanchnic innervation in the adrenal gland is either immature or absent and remains non-functional until the first postnatal week. At birth, hypoxia triggers adrenal catecholamine secretion by a non-neurogenic mechanism that is vital for adapting to the extra-uterine life. The replacement of the non-neurogenic adrenomedullary responses by the neurogenic mechanism is accurately connected to the beginning of the splanchnic secretory-induced nerve function and occurs at the postnatal period [34,35,36]. Interestingly, in rat fetal cells, hypoxia-induced catecholamine release was reported to be shaped by modulating the functioning of calcium-activated potassium (K_Ca_) channels, and ATP-sensitive potassium (K_ATP_) channels [37,38].

The aims of the present study, on cultured rat fetal adrenal medulla chromaffin (AMC) cells, were firstly, to determine the action of gambierol on outward K^+^ currents, using perforated whole-cell voltage-clamp recordings, and secondly, to investigate whether gambierol by itself affects the release of catecholamines using simultaneous current-clamp and single-cell amperometry recordings.

## 2. Results

### 2.1. Effects of Gambierol on Outward K^+^ Current in Rat Fetal Adrenomedullary Chromaffin Cells

In cultured AMC cells, the action of gambierol on K^+^ currents was studied using the perforated whole-cell voltage-clamp configuration. Cells were continuously bathed in a standard physiological solution containing 1 µM tetrodotoxin (TTX) to block the Na_V_ channels, and depolarizing steps (90-ms duration) from a holding potential of −70 mV were delivered to specified membrane potentials. To study the involvement of different K^+^ current components in the total outward K^+^ current, we used the peptide neurotoxins apamin and iberiotoxin which selectively block the small-conductance Ca^2+^-activated K^+^ (SK_Ca_) channels (K_Ca_2.1–2.3, SK1-3 isoforms) [39] and the large-conductance Ca^2+^-activated K^+^ (BK_Ca_) channels (K_Ca_1.1, Slo1) [40], respectively. In addition, glibenclamide was used to target the ATP-sensitive K^+^ (K_ATP_) channel isoforms in neonatal AMC cells [37,41].

As shown on the superimposed recordings of Figure 2(Aa), the addition of apamin (400 nM), iberiotoxin (100 nM) and glibenclamide (200 µM) to the extracellular medium consistently reduced the total outward K^+^ current. Under these conditions, the remaining K^+^ current was further reduced by adding gambierol (100 nM) and was completely inhibited by further addition of the voltage-gated K^+^ channel inhibitors tetraethylammonium chloride (TEA, 10 mM) and 3,4-diaminopyridine (3,4-DAP, 500 µM) to the external solution (Figure 2(Aa)). As depicted in Figure 2(Ba), gambierol (100 nM) only partly inhibited the total K^+^ current when added before the K_Ca_ and K_ATP_ blockers and TEA and 3,4-DAP.

The normalized current-voltage relationships of steady-state K^+^ current amplitudes in the presence of the various agents studied are shown in Figure 2(Ab,Bb). The columns relating the percentage of K^+^ current block induced by the pharmacological agents used, are represented in Figure 2(Ac,Bc). Interestingly, the percentage of K^+^ current inhibition by apamin, iberiotoxin and glibenclamide did not differ significantly in the presence (Δ2 = 58.32 ± 2.82%; *n* = 4), and in the absence (Δ1 = 57.14 ± 4.25%; *n* = 4) of 100 nM gambierol (*p* = 0.8481) (Figure 2(Ac,Bc)).

Taken as a whole, these results indicate that (i) AMC cells are endowed with several types of K^+^ channels contributing to the total outward current, and (ii) gambierol only partly inhibited the total K^+^ current when added after or before the K_Ca_ and K_ATP_ blockers. Therefore, these results strongly suggest that the polyether toxin affects neither the K_Ca_ nor K_ATP_ channels.

Because before the addition of K_Ca_ and K_ATP_ blockers the fraction of the total current blocked by gambierol is too low (approximatively 10%) to allow a proper detection of change in the voltage- and time-dependence of the K^+^ current activation, additional studies, using the perforated whole-cell voltage-clamp configuration, were performed in AMC cells bathed with an external solution containing 1 µM TTX to block the Na_V_ channels, 200 µM glibenclamide to block the K_ATP_ channels, and 1 mM Cd^2+^ to inhibit the K_Ca_ channel activation via Ca^2+^ influx through the Ca_V_ channels. Under these conditions, the voltage-dependence of K^+^ current activation was determined in the absence and presence of gambierol. As shown in Figure 3, the gambierol (100 nM) induced a slight (approximately 5 mV), but significant (*p* = 0.028), negative shift of the voltage-dependence of the K^+^ current activation. Hence, the voltages corresponding to 50% maximal current (V_50%_), before and after gambierol action, were 14.25 ± 0.83 mV (*n* = 4) and 8.93 ± 0.76 mV (*n* = 4), respectively. In addition, the curve slope factor (k) was also slightly, but significantly (*p* = 0.004), higher in the presence than in the absence of the polyether, i.e., 10.17 ± 0.49 mV^−1^ (*n* = 4) and 8.07 ± 0.57 mV^−1^ (*n* = 4), respectively.

The K^+^ current evoked by depolarizing pulses to +40 mV (90 ms duration, from a holding potential of −70 mV) reached a steady-state level (about 40% of control values) within about 4 min after addition of 20 nM gambierol (Figure 4A). Further increase in the gambierol concentration did not promote a supplementary reduction in the outward K^+^-current. When individual K^+^ current traces, recorded every 10 s pulsing, were analyzed before and during the gambierol (20 nM) perfusion, it was clear that gambierol not only reduced the amplitude of the K^+^-current, but also slowed the kinetics of K^+^ current activation by 75.4 ± 10.1% (*n* = 4), with respect to the control (*p* = 0.031) (Figure 4B,C). Hence, before and after gambierol action, the activation time constants of K^+^ current were 3.82 ± 0.39 ms (*n* = 4) and 6.80 ± 1.02 ms (*n* = 4), respectively.

The concentration-response relationship of the gambierol effect on the K^+^ current was established by plotting the steady-state current amplitude, measured in the presence of gambierol (I_G_) and expressed as a percentage of the value obtained before toxin application (I_C_), as a function of the gambierol concentration ([gambierol]). The theoretical concentration-response curve was calculated from a typical sigmoid non-linear regression through data points (correlation coefficient = r^2^), according to the Hill’s equation (using GraphPad Prism v.5 software): I_G_/I_C_ = {(1 − I_ss_)/[1 + ([gambierol]/IC_50_) ^nH^]} + I_ss_, where I_ss_ is the current fraction remaining at high toxin concentrations (Figure 4D, dotted red line), IC_50_ is the toxin concentration necessary to inhibit 50% of the maximal current blocked by gambierol, and n_H_ is the Hill number. Under these conditions, the I_ss_, IC_50_ and n_H_ values were 41.64 ± 0.21%, 5.81 ± 1.56 nM and 0.89 ± 0.32 (r^2^ = 0.982, *n*= 4), respectively (Figure 4D).

### 2.2. Effect of Gambierol on Cathecholamine Release

The use of single-cell amperometry is a valuable quantitative electrochemical method [37,38] to investigate the catecholamine secretion from AMC cells. The simultaneous whole-cell current-clamp and single-cell amperometry combination permitted us to investigate the action of gambierol on catecholamine secretion from fetal AMC cells. Exocytosis events were distinctly detected by positioning a carbon electrode (polarized to +650 mV to allow the oxidation of released catecholamine) as close as possible to the AMC cell, as shown in Figure 5A. Using the current-clamp configuration, the rat fetal AMC cells studied had a mean membrane resting potential of −51.8 ± 3.1 mV (*n* = 32) with a mean coefficient of variation of 0.32 (standard deviation/mean). Eleven of these cells were quiescent and no spontaneous action potential firing was observed while in twenty-one of the cells, spontaneous spike activity was present. In quiescent cells, the gambierol (50 nM) did not change significantly the number of amperometric spikes, related to catecholamine secretion, during 11-s depolarizing current-steps, as illustrated by a representative recording in Figure 5B. It is worth noting that in amperometric recordings, there was a delay between the pulse delivery and the first amperometric spike signal (Figure 5B, lower tracings). Furthermore, there was no correspondence between the recorded action potential (phasic with the current pulse, middle tracings, Figure 5B) and the amperometric signals, in good agreement with previously published data in which single action potentials were ineffective in triggering phasic secretion [42].

It has been previously reported that apamin and iberiotoxin induce catecholamine release in cultured rat fetal AMC cells from F19–F20 fetuses [37]. Therefore, it was of interest to test whether the subsequent addition of apamin (400 nM) and iberiotoxin (100 nM) to the extracellular medium containing the gambierol enhanced the catecholamine secretion. As exemplified in Figure 5B (middle and lower right tracing), during the action of the SK_Ca_ and BK_Ca_ channel blockers (apamin and iberiotoxin, respectively), a significant increase in the frequency of recorded amperometric spike events was detected (Figure 6A), concomitant with a sustained significant membrane depolarization (13.2 ± 0.5 mV, *n* = 3; *p <* 0.05). Further experiments were performed on fetal AMC cells that spontaneously fired action potentials. In those cells, long duration depolarizing current pulses triggered phasic action potentials exhibiting an overshoot of about 30 mV (Figure 5C, middle traces). Under control conditions, such phasic action potentials were followed by repetitive action potentials devoid of overshoot, whose frequency declined (accommodation) during the sustained depolarizing current, as shown in typical recordings (Figure 5C, middle left trace). Under control conditions, the mean action potential frequency during the 11-s current pulse was 2.50 ± 0.62 Hz with a mean coefficient of variation of 0.43 (*n* = 3). The addition of 50 nM gambierol to the medium increased significantly the frequency of action potentials, during the 11-s current pulse, to 4.5 ± 0.22 Hz (*p* = 0.002; *n* = 3) (Figure 5C, middle center trace), when compared to the control, but did not change significantly the amperometric spike events (Figure 5C, lower blue center trace; Figure 6B).

In cells exhibiting repetitive action potentials, blockade of the SK_Ca_ and BK_Ca_ channels by apamin and iberiotoxin, respectively, further enhanced the action potential frequency by about 20% (5.38 ± 0.22 Hz, *p* = 0.023; *n* = 3), and significantly enhanced the amperometric spike events related to catecholamine release (Figure 5C, lower blue right trace; Figure 6B), in a similar manner as in quiescent cells.

On the whole, these results indicate that the specific inhibition of K_V_ channels induced by gambierol does not affect catecholamine release from quiescent or active rat fetal AMC cells. The increased amperometric spike number following the addition of K_Ca_ channel blockers may be the consequence of the increased depolarization induced by the SK_Ca_ and BK_Ca_ channel blockers. Furthermore, the recordings in Figure 5B (lower tracings) clearly show that the release of catecholamines by fetal AMC cells was not dependent on the action potential triggered by the current pulse of long duration but was controlled by the level of membrane depolarization. The frequency of the amperometric events increased when the depolarization of the membrane was larger, probably because more Ca_V_ channels were recruited. In active AMC cells (exhibiting spontaneous action potentials), it is likely that the K^+^ current blocked by gambierol is involved in the control of the action potential firing (Figure 5C, middle center trace).

## 3. Discussion

K^+^ channels constitute an important family of ion channels in excitable neuroendocrine cells and are involved in a number of physiological functions. To the best of our knowledge, this is the first time that the octacyclic polyether toxin gambierol, at nanomolar concentrations, is reported as blocking the K_V_ channels in rat fetal AMC cells. These cells express several types of K^+^ channels contributing to the total outward current, as revealed by using the selective K_Ca_ channel blockers apamin [39] and iberiotoxin [40], and the K_ATP_ channel blocker glibenclamide [41]. Gambierol only partly inhibited the total K^+^ current when added after, or before these channel blockers, suggesting that the polyether affects neither the K_Ca_ nor K_ATP_ channels (Figure 2).

In addition, gambierol slowed the kinetics of K^+^ current activation in fetal AMC cells, implying a delayed opening of the K_V_ channels upon membrane depolarization (Figure 4C). In agreement with previous reports of such an effect [17,43,44], this strongly suggests that the polyether has a greater affinity for the channel resting state [45]. This particularity distinguishes the gambierol action from that of other lipophilic polyether toxins, such as Pacific ciguatoxin-1 (P-CTX-1) which also blocks K_V_ channels in rat myotubes and sensory neurons [46,47]. Interestingly, gambierol action on the K_V_ channels also differs from that of P-CTX-4B (the dinoflagellate-derived precursor of P-CTX-1), which is produced by the same dinoflagellate (*Gambierdiscus toxicus*), and blocks K_V_ channels in myelinated axons, without altering K^+^ current activation [48].

Gambierol and synthetic analogues were previously reported to inhibit K_V_ channels in various cells and tissues, including neurosensory mouse taste cells [17], *Xenopus* skeletal myocytes [18], murine cerebellar neurons [19], human K_V_1.3 channels from T-lymphocytes [22], mammalian K_V_1.1–K_V_1.5 channels expressed in *Xenopus* oocytes [20], and K_V_3.1 channels expressed in both mouse fibroblasts [45] and Chinese hamster ovary (CHO) cells [21]. Gambierol was suggested to affect K_V_ channels by a new mechanism, interacting through a lipid-uncovered binding region of the channel [45]. Electrophysiological work, together with the use of expressed chimeric channels (between K_V_3.1 and K_V_2.1 channels) and homology modelling, revealed that gambierol high-affinity binding occurred in the resting state (when the channel is closed), by disturbing the gate opening and movements of the voltage-sensing domain [23,45,49]. The channel transitions between the resting and the open state require, initially, the dissociation of gambierol that is possible because the polyether has a considerably lower affinity for the open state. In the present work, an approximately 5 mV negative shift of the voltage-dependence of the K^+^ current activation, associated with an increase in the curve slope factor, was detected in the presence of gambierol (Figure 3). These slight modifications in the voltage-dependence of K^+^ current activation are in agreement with the assumption that the polyether is a gating modifier having a putative binding site on K_V_ channels equivalent to that of ciguatoxins, i.e., a cleft between the S5 and S6 segments [23,45,49]. It is worth noting that, in quiescent and active fetal AMC cells, gambierol, in the range of concentrations studied (0.1–100 nM), had no detectable activity on Na_V_ channels, as revealed by action potential recordings, which is in good agreement with previous reports on native [17,18,43] and expressed [20] channels.

The use of simultaneous whole-cell current-clamp and single-cell amperometry allowed for controlling the membrane potential and detecting exocytosis events. Catecholamine secretion in rat fetal AMC cells, lacking splanchnic innervation, is Ca^2+^-dependent [37]. Gambierol did not modify significantly the number of amperometric spike-events triggered by current pulses of long-duration (11 s), causing measurable membrane depolarization that was not significantly different from that of controls (Figure 5B, middle and lower tracings, left and center). The fact that gambierol did not increase catecholamine release in quiescent fetal AMC cells can be explained by the following points: (i) the membrane potential was little affected by the polyether toxin; (ii) during the depolarizing current pulse, the membrane potential was unable to reach the threshold for activating the opening of voltage-gated calcium channels; (iii) other K_V_ channel subtypes remaining unaffected by gambierol, in particular the large-conductance (BK) and small conductance (SK) Ca^2+^-activated K_V_ channel subtypes, curtailed membrane depolarization and voltage-gated Ca^2+^ entry, and therefore catecholamine secretion. Interestingly, block of the K_Ca_ channels, in the continuous presence of gambierol, enhanced membrane depolarization by about 13 mV (Figure 5B, during the 11 s current step), and at the same time, increased significantly the number of exocytotic events related to catecholamine secretion. Such enhanced depolarization is likely to bring the membrane potential above the activation threshold of high-voltage activated Ca_V_ channels, triggering both Ca^2+^ influx and subsequent catecholamine secretion.

In active cells, displaying spontaneous action potentials, gambierol was found to enhance the frequency of action potentials during the 11 s current pulse (Figure 5C) suggesting that the K^+^ current blocked by the polyether may play a role in the control of the action potential firing in fetal AMC cells; however, despite this increase in action potential frequency by the gambierol, no enhancement of the amperometric events was detected, probably because of functional K_Ca_ channels’ activity. The block of the K_Ca_ channels markedly increased the amperometric events, related to catecholamine secretion.

It was surprising to discover that gambierol did not increase catecholamine secretion from rat fetal AMC cells. The action of gambierol at the cellular level depends on the subtype of K_V_ channels that are expressed in a particular cell, their relative proportion, and finally their sensitivity to the polyether toxins. In neurosecretory fetal chromaffin cells, the proportion of K_ATP_ and K_Ca_ channels varied depending on the fetal development stage (F15 or F19–F20) [37,38], and present results. The results obtained in the fetal AMC cells, were quite distinct from previous ones in which gambierol was reported to block a fast K^+^ current in motor nerve terminals, which lengthened the presynaptic action potential duration and thereby increased the amount of Ca^2+^ entry into the terminals and consequently the amount of acetylcholine quanta released upon nerve stimulation [24,25].

Overall, the pharmacological dissection of the several types of K^+^ channels contributing to the total outward current of rat fetal chromaffin cells enhances the knowledge we have on gambierol action, showing that this phycotoxin affects only a fraction of the total K^+^-current component distinct to the K_Ca_ and K_ATP_ currents. Some specific questions related to the type of K^+^ current blocked in fetal AMC cells by gambierol remain unanswered and could motivate forthcoming studies. In addition, our results may help in understanding fetal viability, since gambierol, like other polyether toxins (e.g., brevetoxin-3 and ciguatoxin-1), likely crosses over the mammalian maternofetal barrier. Furthermore, our results show that the K^+^-current block by gambierol in fetal AMC cells lacking splanchnic innervation has no effect on catecholamine secretion, emphasizing the key modulatory role of K_Ca_ currents in controlling exocytosis at this fetal stage (F19–F20).

## 4. Conclusions

In conclusion, (i) several types of K^+^ channels contribute to the total outward current of rat fetal AMC cells; (ii) gambierol only partly inhibits the total K^+^ current when added after or before K_Ca_ and K_ATP_ blockers, and affects neither the K_Ca_ nor K_ATP_ channels; (iii) after blocking the Na_v_ and K_ATP_ channels, and preventing activation of the K_Ca_ channels, gambierol blocks K^+^ currents with a mean IC_50_ of 5.8 nM; (iv) in contrast to ciguatoxins, gambierol slows the kinetics of K^+^-current activation; (v) gambierol does not modify the number of secretory events, related to catecholamine secretion and caused by long-lasting depolarizing pulses; (vi) gambierol increases the frequency of action potentials during a long-lasting current stimulation in cells exhibiting spontaneous action potentials; (vii) surprisingly the Ca^2+^-dependent electrically-elicited catecholamine secretion is not affected by gambierol, but the subsequent block of K_Ca_ channels enhances membrane depolarization, the frequency of action potentials and increases the exocytotic event frequency, highlighting the modulatory role played by K_Ca_ channels in the control of exocytosis from rat fetal AMC cells.

The detailed mechanism of action of gambierol on the various types of transmembrane ion channels, the complexity of ion conductances, and firing activities in excitable rat fetal AMC cells still remain to be further explored.

## 5. Materials and Methods

### 5.1. Chemicals and Toxins

Gambierol was synthesized as described by Fuwa et al. in 2002 [7] and had a purity >97%. The synthetic gambierol was spectroscopically (NMR ^13^C and ^1^H, MS, IR) identical to natural gambierol. Due to the lipophilic nature of gambierol, stock solutions were prepared in dimethyl sulfoxide (DMSO) and diluted with the external physiological solution. The total DMSO concentration in the test solution did not exceed 0.1%. Tetrodotoxin, apamin, iberiotoxin, and glibenclamide were purchased from Alomone Labs (Jerusalem, Israel), and Latoxan (Portes-lès-Valence, France). All other chemicals, including cell culture media, reagents, tetraethylammonium chloride and 3,4-diaminopyridine, were purchased from Sigma-Aldrich (Saint-Quentin-Fallavier, France).

### 5.2. Animals

Adult pregnant Wistar rats were obtained from Elevage Janvier (Le Genest-Saint-Isle, France) and were acclimatized at the animal house for at least 48 h before experiments. Live animals were treated according to the European Community guidelines for laboratory animal handling, and to the guidelines established by the French Council on animal care “Guide for the Care and Use of Laboratory Animals” (EEC86/609 Council Directive; Decree 2001-131). All efforts were made to minimalize animal suffering and to reduce the number of animals used.

Animal care and surgical procedures were performed according to the Directive 2010/63/EU of the European Parliament, which had been approved by the French Ministry of Agriculture. The project was submitted to the French Ethics Committee CEEA (Comité d’Ethique en Expérimentation Animale) and obtained the authorization APAFIS#4111-2016021613253432 v5.

All experiments were performed in accordance with relevant named guidelines and regulations. Pregnant rats were housed in individual cages at constant temperature and a standard light cycle (12 h light/12 h darkness) and had food and water *ad libitum*. On the morning after an overnight breeding, the fetuses were considered to be at day 0.5 of gestation. At day 19 or 20 (F19 or F20), timed-pregnant rats were euthanized by carbon dioxide gas inhalation, and fetuses were rapidly collected and decapitated with scissors according to the guidance of the European Committee DGXI concerning animal experimentation.

### 5.3. Rat Fetal Adrenomedullary Chromaffin Cell Cultures

Fetal adrenomedullary chromaffin cells were obtained from adrenal glands, removed from fetuses collected at F19–F20. The method for chromaffin cell culture is detailed elsewhere [30]. Briefly, the adrenal glands removed from eight to ten rat fetuses were placed in an ice-cold phosphate buffer solution (PBS) to dissect with forceps the capsule and cortex of the adrenal glands under a binocular dissecting microscope.

The isolated medulla was cut into small pieces and treated at 37 °C with 5 mL Ca^2+^-free digestion solution containing collagenase (0.2%, type IA), 0.1% hyaluronidase (type I–S) and 0.02% deoxyribonuclease (type I), to obtain dissociated chromaffin cells. After 30 min of tissue digestion, the enzymatic activity was stopped by adding 400 µL fetal bovine serum.

The digested tissue was rinsed with PBS, three times, and gently grinded with a Pasteur pipette. The cells were re-suspended in 5 mL Dulbecco’s modified Eagle’s medium (DMEM) supplemented with 7.5% fetal bovine serum, 50 IU/mL penicillin and 50 µg/mL streptomycin. Cells were plated onto 35 mm poly-L-lysine-coated dishes (2 mL volume) and kept at 37 °C in a controlled atmosphere (95% CO_2_) for up to 2–4 days before the experiments.

### 5.4. Whole-Cell Voltage- and Current-Clamp Recordings

Membrane currents (under voltage-clamp conditions) and action potentials (under current-clamp conditions in the zero current mode) were recorded from fetal chromaffin cells using the perforated whole-cell technique [50], as previously described [30]. Briefly, recording microelectrodes were pulled from micro-hematocrit capillary tubes with a vertical microelectrode puller (PB-7-Narishige, Tokyo, Japan). Microelectrodes were coated with sticky wax (S.S. White, Gloucester, UK). Pipette resistance had typically 2–5 MΩ when filled with the internal solution. The seal resistance was typically 2–10 GΩ, and about 75–80% of the series resistance (ranging from 12 to 50 MΩ) was compensated electronically, under voltage-clamp conditions. Membrane currents and potentials were monitored with an RK-400 amplifier (Biologic, Claix, France), were filtered at 1–3 kHz (Frequency device, Haverhill, MA, USA), digitized with a DigiData-1200 interface (Axon Instruments, Union city, CA, USA), and stored on the hard disk of a PC computer.

Data acquisition and analyses were performed using the pCLAMP-v.8.0 software (Axon Instruments). The kinetic of K^+^ current activation was determined by measuring the time constant (τ) of the exponential increase, both under control conditions (τ_C_) and in the presence of 20 nM of gambierol (τ_G_). Then, the percentage of change was calculated as [(τ_G_/τ_C_) − 1] × 100. The voltage-dependence of the K^+^ current activation was established by plotting the current (I_K_), expressed as a percentage of its maximal value at +40 mV (I_Kmax_), as a function of membrane potential (V) during 90 ms depolarizing pulses, in the absence and presence of 100 nM of gambierol, as elaborated previously by Hsu et al. in 2017 [44]. The theoretical curve corresponded to data point fits, according to the Boltzmann equation (GraphPad Prism v.5 software): I_K_/I_Kmax_ = 1 − [1/(1 + exp ((V − V_50%_)/k))], where V_50%_ is the voltage corresponding to 50% maximal current, and k is the curve slope factor.

The standard external solution contained (in mM): 135 NaCl, 5 KCl, 2 CaCl_2_, 2 MgCl_2_, 10 glucose, and 10 HEPES (adjusted to pH 7.4 and an osmolarity of 300 mOsm). When necessary, 1 µM tetrodotoxin (TTX) was added to the external solution. The micropipette solution contained (in mM): 105 K^+^ gluconate, 30 KCl, 0.1 CaCl_2_ and 10 HEPES (adjusted to pH 7.2 and an osmolarity of 280 mOsm) and was added with amphotericin-B (24 µg/mL). Neither ATP nor EGTA were added in the internal solution to avoid the catecholamine release being affected, and thus the interpretation of results being complicated, since catecholamine release is calcium-dependent in fetal chromaffin cells. TTX, apamin, iberiotoxin and glibenclamide were added to the external solution. The solutions containing these drugs were freshly made from stock solutions, just before each experiment, and were applied by a custom-made gravity-fed micro-flow perfusion system, positioned as close as possible to the recorded cell. It is worth noting that glibenclamide was reported to have no significant effect on outward K^+^ current under normoxia conditions in neonatal AMCs from P0 rat pups [51]. All experiments were performed at a constant room temperature (21 °C).

### 5.5. Amperometric Recordings from Single Cells

Electrochemical recordings of exocytotic events from single rat fetal AMC cells were performed, as described previously [37]. Cells were visualized with an inverted Olympus microscope. Catecholamine secretion was detected employing 5-μm diameter carbon fiber microelectrodes (purchased from ALA Scientific Instruments, Westbury, NY, USA), and prepared as previously described [49,52].

For the amperometry, a DC potential was applied to the carbon-fiber microelectrode, which appears at the interface between the carbon-fiber and the external solution bathing the cell. If the potential is much greater than the redox potential for a given transmitter, then catecholamines molecules diffuse to the carbon surface and are rapidly oxidized, yielding a current that can be measured. In our experiments, the carbon-fiber microelectrode was positioned adjacent to the individual cell, with the help of a micromanipulator, and a holding voltage of +650 mV was applied between the carbon fiber tip and the Ag/AgCl reference-electrode present in the bath to permit the oxidation of released catecholamines.

Electrochemical currents were filtered at 10 kHz (through a low pass filter) and amplified with a VA-10 current amplifier-system (NPI Electronic GmbH, Tamm, Germany). For peak detection, we used a threshold that was around four times the noise level (around 0.6 pA in our experiments). Amperometric spikes were identified, and carefully inspected offline on a personal computer, using Mini Analysis 5.1 (Synaptosoft, Leonia, NJ, USA) software. All spikes identified by the program were visually examined, and coinciding amperometric events were manually excluded from data sets. Current digitization and storage, data acquisition, as well as the standard external solution, were as described above. All experiments were carried out at a controlled room temperature (21 °C).

### 5.6. Statistics and Data Processing

Data are presented as the mean ± SEM. Comparison between data was completed using a Student’s *t* test. Differences were considered to be statistically significant at *p* < 0.05. The number of experiments (*n*) refers to data obtained from the cells of different rat donors.

## Figures and Tables

**Figure 1 toxins-14-00254-f001:**
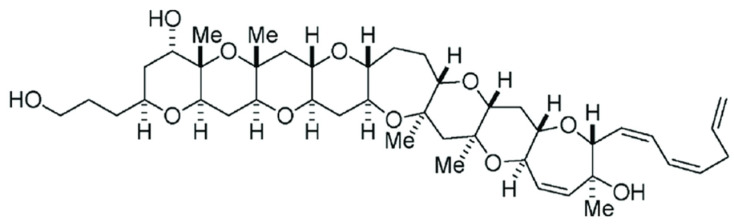
Chemical structure of gambierol.

**Figure 2 toxins-14-00254-f002:**
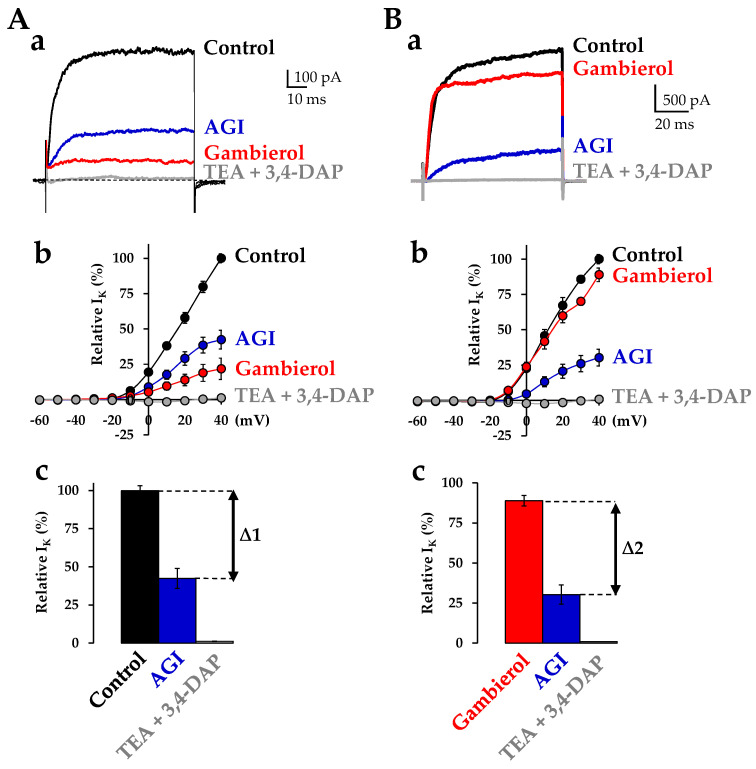
The contribution of various current components (K_Ca_, K_ATP_) and the action of gambierol (100 nM) on total K^+^ current of rat fetal AMC cells. The K^+^ current was recorded using the perforated whole-cell voltage-clamp technique, during 90 ms depolarizing steps from a holding potential of −70 mV, under the conditions indicated in the figure (**A**,**B**). Note in (**A**,**B**) the different sequences of gambierol addition to the external medium. (**a**) Superimposed traces of outward K^+^ currents recorded under control conditions and after the perfusion of the different pharmacological agents indicated. AGI denotes the simultaneous perfusion of 400 nM apamin, 200 µM glibenclamide and 100 nM iberiotoxin. (**b**) Current-voltage relationships. The current was measured at the end of the depolarizing steps and expressed as a percentage of control values for the indicated agents. (**c**) Relative outward K^+^ current contribution (expressed as a percentage of the total current) for the pharmacological treatments indicated. The K^+^ current was measured at the end of the depolarizing steps to +40 mV and expressed as a percentage of control values. Note that the percentage of blocked K_Ca_ and K_ATP_ current components were not significantly different whatever the order of gambierol-induced channel inhibition was. In (**b**,**c**), data represent the mean ± SEM of 4 different experiments.

**Figure 3 toxins-14-00254-f003:**
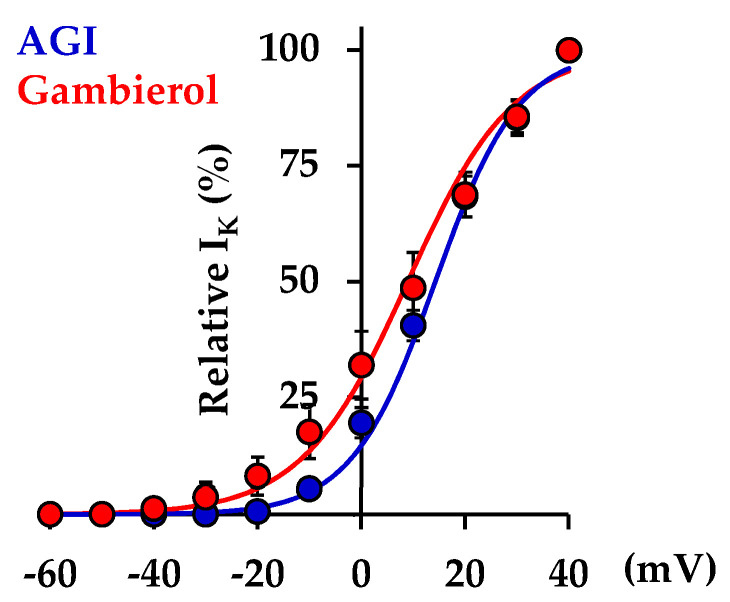
Action of gambierol on the voltage-dependence of K^+^ current activation in rat fetal AMC cells. The current was measured at the end of 90 ms depolarizing steps from a holding potential of −70 mV, expressed as a percentage of its maximal value at +40 mV, and plotted as a function of membrane potential during depolarizing pulses, in the absence (AGI, in blue) and presence (in red) of 100 nM of gambierol. Data represent the mean ± SEM of 4 different experiments. The theoretical curves correspond to data point fits according to the Boltzmann equation, as described in Materials and Methods, with V_50%_ and k values of 14.25 mV and 8.07 mV^−1^ (R^2^ = 0.9989), respectively, for AGI, and 8.93 mV and 10.17 mV^−^^1^ (R^2^ = 0.9997), respectively, in the presence of gambierol. AGI denotes the simultaneous perfusion of 400 nM apamin, 200 µM glibenclamide and 100 nM iberiotoxin.

**Figure 4 toxins-14-00254-f004:**
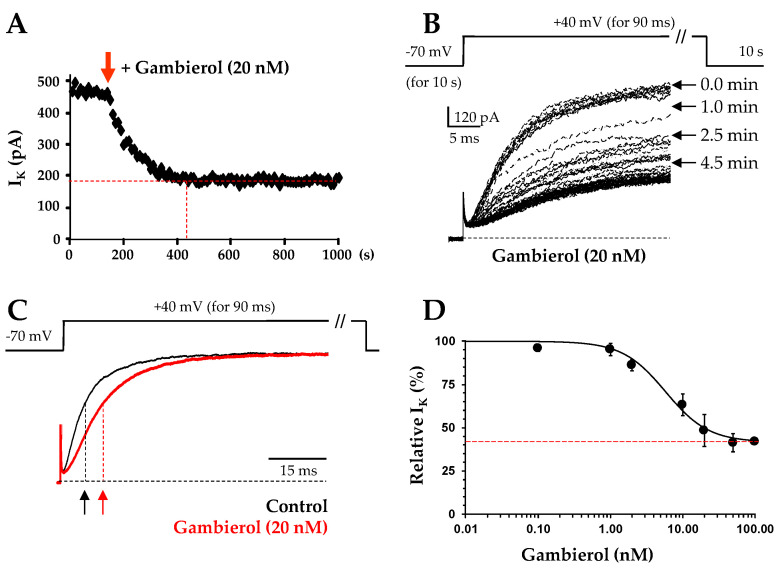
Time-course, kinetics and concentration-dependent action of gambierol on outward K^+^ current in rat fetal AMC cells. (**A**) Time course of 20 nM gambierol action on the K^+^ current measured at the end of 90 ms depolarizing pulses to +40 mV from a holding potential of −70 mV, applied every 10 s (see schema in (**B**)). The red arrow indicates the time of polyether addition to the bath. (**B**) Superimposed K^+^ current recorded every 10-s pulsing, before and after the perfusion of gambierol (20 nM), during 90 ms depolarizing steps to +40 mV from a holding potential of −70 mV (schema). (**C**) Averaged normalized K^+^ current recorded during 90 ms depolarizing steps to +40 mV from a holding potential of −70 mV (schema), under control conditions (black tracing), and after 20 nM gambierol (red tracing), note the slowing of the K^+^ current activation. Data obtained from the same cell. The arrows indicate the activation time constants. (**D**) Concentration-response curve for the effect of gambierol on the steady-state K^+^ current, measured after 90 ms depolarization steps to +40 mV from −70 mV. Each value, determined in the presence of 0.1–100 nM gambierol and normalized to its control value, represents the mean ± SEM of data obtained from 3–4 experiments. The theoretical curve was calculated as described in the text, with I_SS_, IC_50_ and n_H_ values of 42%, 5.8 nM and 0.89 (r^2^ = 0.982), respectively. The external medium in A–D contained 1 µM TTX, 200 µM glibenclamide and 1 mM Cd^2+^ to block, respectively, the Na_V_, K_ATP_ and Ca_V_ channels and prevent, indirectly, the activation of K_Ca_ channels.

**Figure 5 toxins-14-00254-f005:**
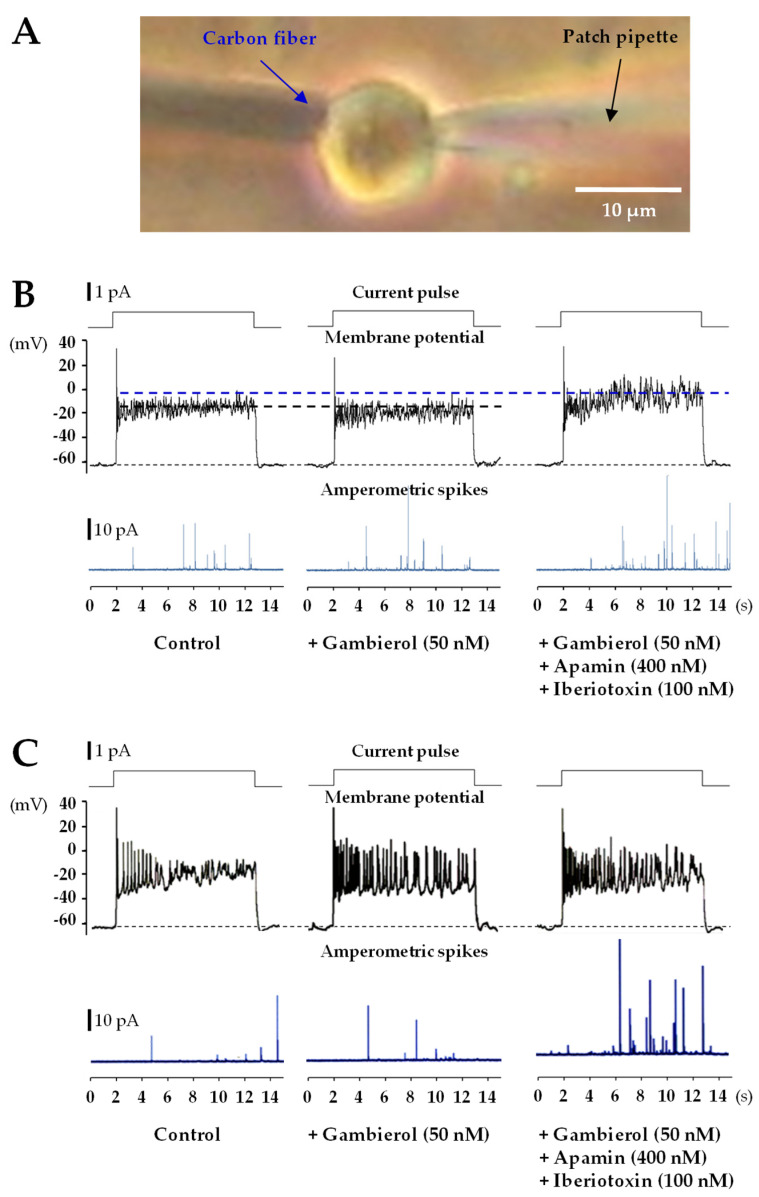
Combined whole-cell current-clamp and amperometric recordings in single rat fetal AMC cells under control conditions, and during the action of gambierol and K_Ca_ blockers. (**A**) Microphotograph of a typical recording configuration for single-cell amperometry, using a polarized carbon fiber electrode to detect exocytotic events, related to catecholamine release, and a whole-cell current-clamp pipette to control the level of membrane depolarization imposed to the cell membrane. (**B**,**C**) Whole-cell current-clamp recordings showing in the upper tracing the current pulse depolarization used to trigger the changes in membrane potential (middle tracings), and the amperometric recording (lower tracings in blue), under control conditions (left), during gambierol action (center), and during the action of gambierol and the K_Ca_ channel blockers indicated (right). (**B**) Example taken from the same quiescent AMC cell (that had no spontaneous action potentials), while the addition of K_Ca_ channel blockers (in the continuous presence of gambierol) enhanced membrane depolarization during the current pulse, by about 14 mV (dotted blue line), and the frequency of exocytotic spikes related to catecholamine secretion events. (**C**) Example taken from an active AMC cell (that had spontaneous action potentials). Note that gambierol, as well as the addition of K_Ca_ channel blockers (in the continuous presence of gambierol), enhanced the frequency of repetitive action potentials during the current pulse depolarization, with respect to the control. Note also, that the frequency of exocytotic spikes, related to catecholamine secretion events, was only increased after the addition of K_Ca_ channel blockers.

**Figure 6 toxins-14-00254-f006:**
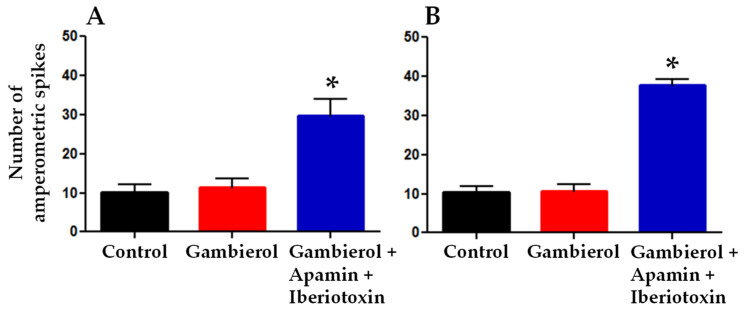
Number of amperometric spikes related to catecholamine release under control conditions, and during gambierol (50 nM), apamin (400 nM) and iberiotoxin (100 nM) applied cumulatively to the external medium by perfusion. Data obtained from fetal AMC cells showing initially either no action potential firing (**A**) or spontaneous spike activity (**B**), during stimulation with depolarizing current pulses of 11-s duration. Each column represents the mean ± SEM of 3 different experiments. Note that gambierol did not modify the number of release events, while a significant 2.7-fold (**A**) and 3.5-fold (**B**) increase occurred after the addition of K_Ca_ blockers. *: *p* ≤ 0.005 versus gambierol.

## Data Availability

Data is contained within the article.

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
