# Peer review of "Gambierol Blocks a K+ Current Fraction without Affecting Catecholamine Release in Rat Fetal Adrenomedullary Cultured Chromaffin Cells"

_toxins, 2022, doi:10.3390/toxins14040254_

Round 1
Reviewer 1 Report
see attached.

Reviewer 2 Report
The authors aimed to determine the type of voltage-gated potassium channel that gambierol blocks. For this purpose, by using diverse pharmacological probes, the authors excluded a series of K+-ion channels subtypes and conclude that gambierol is not a potent cathecolamine-releasing toxin and Ca2+-dependent K+-channel is the main player for the release of this mediator in chromaffin cells.
Based on the data that were presented, the manuscript is acceptable.
Reviewer's recommendations:
Preferentially, too add a figure of the structural formula of gambierol and related molecules.
Also, important to add a table in which are listed the main ion channels types that are sensitive (or not) to gambierol (and analogs) exert its effect.
Reviewer 3 Report
The article presents a small study of the effect of octacyclic polyether toxin gambierol on potassium currents of cultured rat fetal adrenal medulla chromaffin (AMC) cells. It is clearly shown that the toxin inhibits one component of total potassium current at nanomolar concentrations and this component does not belong to Ca- or ATP-activated K-channels. It has also been shown that the toxin does not cause the release of catecholamines in these cells. The methods are clearly described, the results are described in detail and analyzed.
Unfortunately, the question of which Kv channels are present in AMC cells and blocked by the toxin has not been clarified and discussed. It would be nice to provide data on the concentrations of gambierol in which it blocks various subtypes of voltage-gated potassium channels. Are any data that gambierol change the kinetics of Kv current activation?
Minor remarks
1- Figure 1 and its signature - Panels A, C, D - correspond to one scheme of gambierol addition (after KCa and KATP blockers,) B, E, F - to another one (before KCa and KATP blockers). This should be more clearly reflected in the figure and its caption, perhaps merge pictures that match the same pattern
2 - Fig 3B, right panel - it is not clear that this is a recording in the continuous presence of gambierol as noted in the caption and in the text, The same applies to Figure 4.
- lines 12-13 - an unsuccessfully formulated sentence, IC 50 is not a characteristic of a current, but a toxin
- lines 39-41 – “included” is redundant
- line 42 – “ the” first
- lines 80-82 - replace “First” with “firstly” and “second’ with “secondly” or use (1) and (2)
Round 2
Reviewer 1 Report
No further comments are provided in the revised manuscript.